# Rethinking supervised learning: insights from biological learning and from calling it by its name

**Alex Hernandez-Garcia**[*]
Institute of Cognitive Science, University of Osnabrück, Germany
Max Planck School of Cognition
`ahernandez@uos.de`

## Abstract

The renaissance of artificial neural networks was catalysed by the success of classification models, tagged by the community with the broader term *supervised learning*. The extraordinary results gave rise to a hype loaded with ambitious promises and overstatements. Soon the community realised that the success owed much to the availability of thousands of labelled examples. And *supervised learning* went, for many, from glory to shame. Some criticised deep learning as a whole and others proclaimed that the way forward had to be "alternatives" to supervised learning: *predictive*, *unsupervised*, *semi-supervised* and, more recently, *self-supervised learning*. However, these seem all brand names, rather than actual categories of a theoretically grounded taxonomy. Moreover, the call to banish supervised learning was motivated by the questionable claim that humans learn with little or no supervision. Here, we review insights about learning and supervision in nature, revisit the notion that learning is *not* possible without supervision and argue that we will make better progress if we just call it by its name.

## 1 Introduction

The re-emergence of deep learning during the last decade due to the noteworthy achievements of artificial neural networks (ANN) built up a sort of philosophy that nearly anything could be automatically learnt from data without human intervention, in contrast to the previous approaches: "[hand designing good feature extractors, engineering skill and domain expertise] can all be avoided if good features can be learned automatically using a general-purpose learning procedure. This is the key advantage of deep learning" (LeCun et al., 2015).

However, this claim was overstated and can be misleading, since it misses the fact that the success of deep learning has required iterative hand design of network architectures that demanded high and collective engineering skill and large doses of interdisciplinary domain expertise. Additionally, deep learning owes much to the immense computational power poured into training artificial networks (Amodei & Hernandez, 2018; Schwartz et al., 2019) and to the human effort of manually labelling thousands of images and other data modalities (Russakovsky et al., 2015; Cao et al., 2018).

At the same time, the need for much data has been used for motivating new research avenues and goals of deep learning that can be equally misleading and overly ambitious: "learning a class from a single labelled example", based on the statement that "humans learn new concepts with very little supervision, [but] the standard supervised deep learning paradigm does not offer a satisfactory solution for learning new concepts rapidly from little data" (Vinyals et al., 2016). Some of the work that followed up these proposals certainly brought about promising methods and insights, but at the

---

[*]Personal webpage: alexhernandezgarcia.github.io

price of a publication landscape full of confusing terminology and tangled research directions that are hard to navigate.

Here, we reflect upon fundamental concepts in machine learning such as supervision and inductive biases, which in spite of resting on solid theoretical grounds, are at the core of misconceptions and overstatements about deep learning commonly seen in the literature. We revisit insights from biological learning and from statistical learning theory with the aim of tempering certain claims and promises of deep learning, mitigating the confusion over the terminology and clarifying the relationships between different research directions.

## 2    Rethinking supervised learning

The realisation that the success of deep learning was largely due to the availability of huge labelled data sets prompted various reactions: some strongly questioned the usefulness of the algorithms (Marcus, 2018); some delved into the question of whether neural networks generalise beyond or simply memorise the training examples (Zhang et al., 2017; Arpit et al., 2017); and some opened new research horizons arguing that models should generalise from a few examples (Vinyals et al., 2016). As a consequence, multiple research programmes, with various brands, followed up with the aim of minimising or removing the need for "supervision" to train neural networks: few-shot, one-shot, zero-shot learning; predictive, unsupervised, semi-supervised, self-supervised learning, and so on.

Exploring alternatives to classification and improving the efficiency of learning algorithms should be a priority, and related approaches had been subject of machine learning research before the explosion of deep learning (Hinton & Sejnowski, 1999; Chapelle et al., 2006). However, the current publication and discussion trends in the field denote overambitious promises, based partly on misconceptions and overstatements about learning in nature, and amplified by overselling nomenclature. While much of the research output does provide us with useful techniques and insight, the confusion amidst the hype run the risk of leading many astray.

In this section we aim to reassess these promises by first drawing insights from biological learning in the hope for setting upper bounds—at least in the near future—to our expectations about deep learning; and subsequently rethink the terminology around supervised learning commonly employed in today's deep learning literature.

### 2.1    Supervised biological learning

The link between artificial intelligence—specifically artificial neural networks (Rosenblatt, 1958; Fukushima & Miyake, 1982)—and biological learning systems is intrinsic to the field, as one long-term goal of artificial intelligence is to mirror the capabilities of human intelligence. However, these capabilities are sometimes overestimated. One example is the argument that intelligence in nature evolves without supervision and that humans and other animals learn—to visually categorise objects, for example—with little or no supervision (Vinyals et al., 2016; Marcus, 2018; Morgenstern et al., 2019). In what follows, we will discuss three aspects of biological learning to argue against this view, in order to gain insights that better inform our progress in machine learning: first, we will discuss how generalisation requires exposure to relevant *training data*; second, the role of evolution and brain development; third, the variety of supervised signals that the brain has access to.

**Generalisation requires exposure to relevant training data**    First, in the argument that machine learning models should generalise from a few examples, there seems to be a promise or aspiration that future better methods will be able to perform robust visual object categorisation among many object classes after being trained on one or a few examples per class. While a primary objective is to develop techniques that efficiently extract the maximum possible information from the available examples, we should also remind ourselves that no machine learning algorithm can robustly learn anything that cannot be inferred from the data it has been trained on. Although this may seem to contradict certain current trends and statements in the literature, we should also bear in mind that learning in nature is not too different.

Even though the human visual system, for example, is remarkably robust, its capabilities are optimised for the tasks it needs to perform and largely shaped by experience, that is the *training data*—and years of evolution, as we will discuss below. For instance, from the literature on human visual perception,

we know that object recognition is sensitive to changes in view angle (Tarr et al., 1998), even though largely invariant under certain conditions (Biederman & Bar, 1999). A well-studied property of human vision is that our face recognition ability is severely impaired if faces are presented upside down (Yin, 1969; Valentine, 1988). Setting aside the specific complexity of face processing in the brain, a compelling explanation for this impairment is that we are simply not used to seeing and recognising inverted faces. More generally, human perception of objects and our recognition ability is greatly affected when we see objects from unfamiliar viewpoints (Edelman & Bülthoff, 1992; Bülthoff & Newell, 2006; Milivojevic, 2012).

Furthermore, although better than the *one-shot* or *few-shot* generalisation of current ANNs, humans also have limited ability to recognise novel classes (Morgenstern et al., 2019). Interestingly, experiments with certain novel classes of objects known as Greebles showed that with sufficient training, humans can acquire expertise in recognising new objects from different viewpoints, even making use of an area of the brain—the fusiform face area—that typically responds strongly with face stimuli (Gauthier et al., 1999). This provides evidence that generalisation to multiple viewpoints is possible but only developed after exposure to similar conditions.

**The role of evolution and brain development**    Second, the commonplace comparison of artificial neural networks with the brain often misses a fundamental component of biology, recently brought to the fore by Zador (2019) and Hasson et al. (2020), although considered since the early days of artificial intelligence (Turing, 1968): the role that millions of years of evolution have played in developing the nervous systems of organisms in nature, including the human brain.

The properties and capabilities of ANNs trained for visual object categorisation are often contrasted with those of the adult brain (Yamins et al., 2014; Khaligh-Razavi & Kriegeskorte, 2014; Güçlü & van Gerven, 2015; Rajalingham et al., 2018; Geirhos et al., 2018). However, the artificial neural network models used for these comparisons are typically trained from scratch[2], from random initialisation, to learn how to categorise its first visual categories. In contrast, a large part of the brain connectivity is encoded genetically and certain properties of the visual cortex are known to be innate, that is developed without prior exposure to visual stimuli (Zador, 2019).

While comparing ANNs with the adult brain certainly gives us interesting insights, if our goals are to construct better models of the brain and to understand the mechanisms of learning algorithms, it seems reasonable to also study the role of evolution and brain development in the infant brain. In fact, the process of training a neural network on simple tasks from random initialisation may be more similar to brain evolution and learning in the infant brain (Harwerth et al., 1986; Atkinson, 2002; Gelman & Meyer, 2011), than to learning in the adult brain. This may be another reason to temper the expectations that neural networks be able to generalise robustly from a few examples, without *hard-wiring* some of the innate properties of the brain (Lindsey et al., 2019; Malhotra et al., 2020) or, alternatively, simulating part of the evolutionary process.

**Supervised signals in the brain**    Third, another commonly found argument has it that children, and other animals in general, learn robust object recognition without supervision: "a child can generalize the concept of 'giraffe' from a single picture in a book" (Vinyals et al., 2016). First of all, we should recall again the role of evolution, which can be interpreted as a pre-trained model, optimised through millions of years of data with natural selection as a supervised signal (Zador, 2019). Second, we will argue against the very claim that children learn in fully unsupervised fashion.

Obviously, the kind of supervision that humans make use of is not that of classification algorithms: we do not see a class label on top of every object we look at. However, we receive supervision from multiple sources. Even though not for every visual stimulus, children do frequently receive information about the object classes they see—for instance, parents would point at objects and name them, then we learn how to read, and so on. Furthermore, humans usually follow a guided hierarchical learning: children do not directly learn to tell apart breeds of dogs, but rather start with umbrella terms and then progressively learn down the class hierarchy (Bornstein & Arterberry, 2010). Hasson et al. (2020) mention other examples of supervision from *social cues*, that is from other humans, such

---

[2]Some interesting and promising areas in machine learning research deviate from this standard approach. For example, transfer learning and domain adaptation study the potential of features learnt on one task to be reused in different, related tasks (Zhuang et al., 2019), and continual learning studies the ways in which machine learning models can indefinitely sustain the acquisition of new knowledge without detriment of the previously learnt tasks (Mundt et al., 2020). These approaches are inspired by biological learning or share interesting properties with it.

as learning to recognise individual faces, produce grammatical sentences, read and write; as well as from embodiment and action, such as learning to balance the body while walking or grasping objects. In all these actions, we can identify a supervised signal that surely influences learning in the brain (Shapiro, 2012).

Moreover, besides this kind of explicit supervision, the brain certainly makes use of more subtle, implicit supervised signals, such as temporal stability (Becker, 1999; Wyss et al., 2003): The light that enters the retina is not a random signal from a sequence of rapidly changing arbitrary photos, but a highly coherent and regular flow of slowly changing stimuli, especially at the higher, semantical level (Kording et al., 2004). At the very least, this is how we perceive it and if such a smooth perception is a consequence rather than a cause, then it should be, again, a by-product of a long process of evolution.

In the light of these insights from evolutionary theory and the explicit and implicit supervision that drives biological learning, we argue that we should temper the claims and aspirations that artificial neural networks should learn without supervision and from very few examples. Instead, we may benefit from rethinking the concept of supervision, embrace it and try to incorporate the forms of supervision present in nature into machine learning algorithms.

## 2.2   Supervised machine learning

If we open a machine learning textbook (Murphy, 2012; Abu-Mostafa et al., 2012; Goodfellow et al., 2016), we will most surely find a taxonomy of learning algorithms with a clear distinction between *supervised* and *unsupervised* learning. However, while this separation can be useful, the boundaries are certainly not clear. As a matter of fact, strictly speaking, unsupervised learning is an illusion. If we take a look at the deep learning literature of the past years, we will also find abundant work on some variants supposedly *in between*—semi-supervised learning, self-supervised learning, etc.—whose definitions are all but clear.

**Catastrophic forgetting of old concepts**   If we recall a classical result in statistical learning theory and inference, the *no free lunch* theorem (Wolpert, 1996), no learning algorithm is better than any other at classifying unobserved data points, when averaged over all possible data distributions. Therefore, we need to constrain the distributions or, in other words, introduce prior knowledge—that is *supervision*. Recently, Locatello et al. (2018) obtained a related result for the case of unsupervised learning of disentangled representations: without inductive biases for both the models and the data sets, unsupervised disentanglement learning is impossible. These results are purely theoretical and have limited impact on real world applications Giraud-Carrier & Provost (2005), precisely because in practice we use multiple inductive biases, even when we do so-called unsupervised learning.

In a strict sense, even the classical, *purely* unsupervised methods, such as independent component analysis or nearest neighbours classifiers, make use of inductive biases, such as independence or minimum distance, respectively. Without inductive bias, learning is not possible. Although this is not news, the terminology used in the machine learning literature seems to neglect these nuances and evidences that the field suffers catastrophic forgetting of well-established notions.

**The brands of *alt-supervised* learning**   Particularly in deep learning and computer vision, the term *supervised* learning is often used to actually refer to *classification*, that is models trained on examples labelled according to, for instance, the object classes. In turn, *unsupervised* learning is used for any model that does not use the labels, regardless of what other kind of supervision it may use. Further, the term *semi-supervised* learning refers in practice to models that are trained with a fraction of the labels, but are tested on the same classification benchmarks. More recently, the term *self-supervised* learning has gained much popularity[3], referring to models that are trained on tasks other than the standard task defined by classification labels.

**Supervision comes in different flavours**   Some of the methods proposed under these brand names are certainly useful—that is not the subject of criticism of this work—but the terminology is overly confusing and unnecessary. First, there are not consensus definitions of these categories. A newcomer

---

[3]Especially, after Yann LeCun's keynote presentation at ISSCC in February 2019. In December 2016, he had titled his NeurIPS keynote presentation "Predictive Learning", to refer to similar ideas, pointing out that "that's what many people mean by unsupervised learning". Later, he wrote: "I Now call it 'self-supervised learning', because 'unsupervised' is both a loaded and confusing term".

would easily fall into a scientific rabbit hole trying to tell these terms apart through publications—not to mention if they incorporated social media discussions into their endeavour. Second, from a theoretical point of view, both the conventional classification models and the recent wave of "self-supervised" tasks can all be formalised within the category of supervised learning. Only that supervision comes in different flavours, not only as classification labels.

We have seen examples of different forms of supervision used by humans and other animals. In machine learning, the field focused for many years on a few loss functions, such as classification and simple forms of regression. The relatively recent explosion of deep learning has brought about the development of several libraries for automatic differentiation (Baydin et al., 2017), which in turn have enabled the proposal of multiple loss functions with various types of supervision (see (Jing & Tian, 2020) for a recent review) that can easily be optimised numerically by stochastic gradient descent and artificial neural networks. This opens in fact a great opportunity to incorporate new inductive biases from biological learning into machine learning algorithms, beyond classification labels. However, we contend that this is also supervised learning and we should call it by its name.

The terms *self-* and *semi-*supervised imply that the methods use *less* supervision, but, in fact, they should have been called, if anything, *hyper-supervised*[4] learning. Pure classifiers use the labels as main supervised signal, but self- and semi-supervised methods often make use of a wide range of surrogate tasks with supervised signals defined by humans. A well-known example is image data augmentation: Although at first glance it may seem a naïve technique, it actually encodes rich prior knowledge about visual perception and that is why it outperforms explicit regularisation methods with less effective inductive biases Hernández-García & König (2018) and is widely used in "semi-supervised" tasks Laine & Aila (2016).

## 3 Discussion

In this paper, we have first discussed some of the overambitious promises and misconceptions of the deep learning hype around the need for human intervention and supervision in machine learning. In particular, we have critically reviewed the various terms that are currently used to refer to supposed alternatives to supervised learning: semi-, self- and unsupervised learning, among others. In sum, we pointed out that all these approaches are in fact supervised learning—though not necessarily classification—and the machine learning (research) community would benefit from using more rigorous, less overselling nomenclature.

Supervision is not evil. It is at the core of statistical learning theory: learning is impossible without inductive biases or supervision. But it has different flavours, not only classification. Neither is deep learning some sort of exceptional solution to learn without human intervention and supervision, nor is it a hopeless model class because it requires large data sets (Marcus, 2018). The human visual system is exposed to a lot of stimuli too. One exceptional advantage of deep learning is precisely that it is possible to effectively optimise different learning objectives, almost end-to-end, from large collections of nearly naturalistic sensory signals, such as digital images (Saxe et al., 2020). While other models are known to scale poorly as the amount of data increases, neural networks excel at fitting the training data and interpolating on unseen examples (Belkin et al., 2019; Hasson et al., 2020). This is a feature, not a bug. But we will make better progress if we exploit these advantages of deep learning without neglecting that supervision will always be necessary—the critical goal is how to best incorporate it and exploit it.

In this regard, we can argue that deep learning needs *more supervision*, and not less. A major focus of the deep learning community in the last decade has been image object classification. This has brought about unprecedented progress and unveiled the limitations of having classification as chief task and class labels as main supervised signal. For example, deep classifiers have been found to learn spurious features that are highly discriminative for the classification task but with little true generalisation power and clearly not aligned with perceptual features (Jo & Bengio, 2017; Wang et al., 2019; Geirhos et al., 2020). In fact, this mismatch has been argued to be at the root of adversarial vulnerability (Ilyas et al., 2019) and seems to be the consequence of training highly expressive, over-parameterised models in heavily unconstrained tasks. This can be addressed with meaningful constraints, that is more and richer supervision, possibly inspired by human perception and biological learning. For example, combining a classification loss with a similarity loss inspired by the invariance in the visual

---

[4]The authors explicitly discourage the addition of a new term to the already too confusing list.

cortex yields more robust representations without detriment to categorisation (Hernández-García et al., 2019). Expanding in this direction leads to multi-task learning and representation learning, away from simply classification.

We have drawn parallels from cognitive science and human perception to contend that learning in nature also requires supervision, where it too has multiple forms. Even evolution can be regarded as an optimisation process where natural selection is the supervised signal (Zador, 2019). We have argued, as others have before, that these insights from biology and neuroscience offer a great opportunity for machine learning research to draw inspiration and calibrate its compass.

Finally, we recall that most of the learning theory has been developed for simple loss functions such as binary classification or mean squared error regression, but certain methods successfully used in practice today escape the available theory. Given the success of this kind of more complex supervised objectives and their connection with perception, the study of these methods from a theoretical point of view might be a fruitful direction for future work.

## Broader Impact

Since this article does not present a new method or results from data sets, potential risks of "bias in the data" or "failure of the system" do not apply. As a critical review of current trends in the field, some researchers could potentially be affected depending on their work. We declare that we do not intend to negatively affect any individual researcher, but rather potentially improve scientific value through a constructive reflection.

## Acknowledgments and Disclosure of Funding

This project has received funding from the European Union's Horizon 2020 research and innovation programme under the Marie Sklodowska-Curie grant agreement No 641805. Supported by BMBF and Max Planck Society.

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
