# OpenReview forum: "Rethinking supervised learning: insights from biological learning and from calling it by its name"
_NeurIPS.cc/2020/Workshop/SVRHM — SVRHM@NeurIPS Poster_

### Official Review · AnonReviewer1 · 2020-10-20
**Review of "Rethinking supervised learning: insights from biological learning and from calling it by its name"**

**Rating:** 6
**Confidence:** 4

**Review:**

# Review of "Rethinking supervised learning: insights from biological learning and from calling it by its name"

## Summary & recommendation
The paper "Rethinking supervised learning: insights from biological learning and from calling it by its name" provides a critical perspective on the differences between supervised and unsupervised learning. The authors argue that "unsupervised learning" is a paradox: "learning is impossible without inductive biases or supervision", hence "unsupervised learning" is not possible. Instead, the authors argue that the many flavours of what the field calls "unsupervised" or "self-supervised" learning are forms of supervised learning in disguise (for instance: the forms of data augmentation during self-supervised learning implicitly incorporate knowledge about visual perception, such as the requirement that object identity should not change under certain augmentations). Hence, the authors argue that the field should start calling it "by its name" - supervised learning.

Overall, I enjoyed reading this article. Not because I agree with everything, not because I think everything is presented in the most accurate and fair fashion, but because the article is thought-provoking, and the field can only benefit from such kind of articles. Hence, if the workshop organisers think that a perspective article (without experiments) is within the scope of the workshop, I believe it might lead to some interesting and important discussions.

I am listing a few points where I believe the paper could improve.

### 1. Descriptive without being prescriptive
The authors see a "publication landscape full of confusing terminology and research directions". I personally am inclined to agree with the first point (confusing terminology) and to disagree with the second one (confusing research directions). In either case, the authors set themselves the aim of "mitigating the confusion over the terminology and research directions", and it currently falls short from achieving this ambitious aim. The authors believe that the problem originates because the boundaries between various terms are unclear (e.g. predictive learning / self-supervised learning / semi-supervised learning / unsupervised learning). However, do we really achieve clarity if we simply put everything into one pot by using the same term, "supervised learning", for all of it? Ultimately, the paper could achieve the biggest impact if the authors would propose a meaningful, useful taxonomy rather than "discourage the addition of a new term to the already too confusing list". The degree of supervision could be one dimension in this taxonomy, and perhaps the form of supervision another: whether supervision comes in the form of labels, data augmentation, and so on.

### 2. Supervised biological learning section
Overall, I found section 2.2 really nice (supervised ML) and section 2.1 a bit confusing at times. I list a few points of confusion here:
- "evolution, which can be interpreted as a pre-trained model". This is not how evolution would usually be interpreted - a pre-trained model can directly be used to solve high-level tasks like categorisation, and no human baby would be able to categorise many objects just days after being born.
- The authors mention that humans learn from evolution and during development. They conclude from this that we should therefore "temper the expectations that NNs be able to generalise robustly from a few examples". Logically, this statement does not follow for me. Just because humans learn from evolution and during development doesn't automatically mean that machines will need to go through the same process. There seems to be the implicit assumption that the human/biological way is the only way (which is reminiscent of anthropomorphism).
- "These comparisons, although interesting, implicitly assume that the process by which a neural network
learns its first visual categories from random initialisation is equivalent to the development of a brain
from birth to maturity." This statement does not properly reflect the papers to which it refers. Some of the cited papers don't assume anything in this regard but instead compare biological to artificial vision in an open-ended fashion.


### 3. Acknowledging insights from within the ML community
The paper at points conveys the impression of an "outsider" / "bystander" looking at the field of ML, and bestowing insights upon the field from the outside. It would only be fair to acknowledge those within ML a bit more who are pointing out similar issues. Just to give an example, Aaron van den Oord recently gave a talk at an ECCV workshop on self-supervised learning, and mentioned quite a few aspects in which current "self-supervised" methods use supervision (e.g. curated datasets, hyperparameter optimisation). Talk link: https://www.youtube.com/watch?v=jJozjCG8Cqs&ab_channel=OxfordVGG. I'm sure there are more positive examples, and a balanced discussion would benefit from a few more positive examples from within ML.

### 4. Toning down language
While I am sympathetic towards the authors' apparent dislike for "brand names" and I also appreciate their critical look at the field, I have the impression that some statements are unnecessarily sharp and could be toned down a little. For instance, "publication landscape full of confusing terminology and research directions", "overselling nomenclature", "overambitious promises and misconceptions of the deep learning hype". In combination with a lack of acknowledgment of insights from within the ML community, this might put some readers off. The paper's core arguments can only gain from using more nuanced language, because unnecessary sharpness might just be a distraction from the content.


### 5. Visualise key point(s) in a figure?
This is just a suggestion. When reading the paper, it occurred to me that it might be helpful to summarise the key results in a figure. For instance, a figure that visualises the typical building blocks (training scheme, architecture, dataset) of what the field currently calls "self-supervised learning", and then point out visually where the supervision "creeps in": choice of data augmentation (knowledge about human visual perception), choice of test distribution (no free lunch), optimisation (evolution), ...

---

> ### Public Comment · ~Alex_Hernandez-Garcia1 · 2020-12-03
> **Responses to reviewer's comments and description of changes**
>
> I would like to thank you for your detailed and insightful review. I am glad you enjoy reading the article and found it thought-provoking. I am grateful too for the multiple suggestions for improvement. They have been very useful to improve the updated manuscript. I will briefly address your comments and describe the relevant changes to the manuscript.
>
> _1. Descriptive without being prescriptive_
>
> I essentially agree with this general concern and I believe that it would be good to propose an alternative, more rigorous taxonomy. However, that is a challenging endeavour, which should arise from extensive, deep discussion within the community. This paper addresses the first step, which is identifying and describing a problem. I am looking forward to discussing this at the workshop and get a better idea of the sentiment towards this criticism and the need for a better taxonomy.
>
> We do think that identifying and describing some problems of the current terminology contributes to mitigate the confusion, but we agree that the paper does not primarily propose an alternative taxonomy. Therefore, I have changed to wording in the abstract and introduction to relax and clarify the claims in that regard.
>
> _2. Supervised biological learning section_
>
> Regarding the issue of evolution, I draw significant inspiration from Zador (2019) and Hasson et al. (2020). In their papers it is discussed in detail how evolution can be interpreted as a pre-trained model and some innate capacities of living organisms, including humans.
>
> I agree that the human/biological way is not the only way. To me, it is the best way we know so far though, but that is a different discussion. In any case, the complete logic here is the following: some authors do overstate some capabilities of biological learning to motivate research goals (as briefly review in the paper). If we use biological leanring as motivation, then it does make sense to say that we should temper the expectations, in my opinion.
>
> I have rewritten the section "The role of evolution and brain development", based on your comments, to hopefully make it less confusing and better reflect to intended goal.
>
> _3. Acknowledging insights from within the ML community_
>
> The intention was not to give the point of view of an "outsider" to the field of machine learning. In fact, I am personally quite in the field. I have rewritten some sentences to make clear that I acknowledge the usefulness and potential of the progress made in deep learning. However, I believe that the positive aspects of the field are quite well-known. Therefore, in this short paper I decided to focus on some points of criticism with a constructive intention.
>
> _4. Toning down language_
>
> I have also reviewed the tone of several sentences based on your feedback, although I have kept some relatively sharp and perhaps unusual terms. This was deliberate and is intented to provoke critical thoughts in the reader, though I am aware it may not work with all readers.
>
> _5. Visualise key point(s) in a figure?_
>
> This is a great idea, thanks! However, I have not yet come up with a figure I am happy to include. I keep thinking and working on this and I will hopefully add it in a future version of the paper.

---

### Official Review · AnonReviewer3 · 2020-10-28

**Rating:** 7
**Confidence:** 4

**Review:**

1. Summary

The paper presents an in-depth discussion of the term supervised learning. The authors argue that (currently) popular terms in machine learning research such as unsupervised, self-supervised, semi-supervised, one-shot, few-shot, zero-shot learning etc. are confusing as they don't correspond well to the types of learning we know from humans and other biological systems. One key point is that human learning is based on a lot of supervision however little of which is presented in the form of explicit labels as commonly used in classification.


2. Strengths/Weaknesses
+ Very relevant topic/discussion
+ The paper nicely contrasts terms and insights from human and computer learning pointing out conceptual problems that arise from these differences
+ To me the selection and presentation of cited papers looks pretty good
- There is little to dislike as I think this paper contributes some important points to the discussion around what we can expect machine learning models. If anything I felt the structure was not perfect yet with the separation into supervised biological and machine learning. As the whole paper revolves around this contrast it may have made more sense to structure the paper according to certain points like the influence of evolution, the diversity of cues humans use as labels, the expectation that DNNs perform like human adults, the discussion of no free lunch etc.

3. Recommendation

I recommend accepting this paper as it adds important input and context to current discussions what we can expect from machine learning models.

4. Comments
- The authors correctly point out that expecting DNNs to learn concepts from a single example is kind of misleading. It thus may be interesting to mention that the current leader in such few-shot learning tasks ("Big Transfer", Koleshnikov et. al. 2019) is a model built for transfer learning. It uses extensive pre-training to do one-shot ImageNet and basically performs this task by using it's existing knowledge to order the next examples.
- It would be great to relate the presented view to "Shortcut Learning" (Geirhos et. al. 2020) as it seems to contain an essential clue why DNNs so often learn something completely different from what we expect: The type of information used for supervision of biological and machine learning systems are very different!

---

> ### Public Comment · ~Alex_Hernandez-Garcia1 · 2020-12-03
> **Responses to reviewer's comments and description of changes**
>
> I would like to kindly thank you for the detailed review. It was encouraging to read that you think the paper can potentially add important input to the meta-discussion within machine learning. Also your comments have certainly contributed to the updated manuscript. Below I will briefly repply to the most important comments of the review and describe some changes.
>
> The suggestion about an alternative structure for the paper made me rethink some arguments of the paper, but since this change would imply major rewriting of the paper, I decided to keep the original structure and add only smaller scale changes.
>
> Already in the original version we briefly discussed in a footnote that transfer learning (and continual learning) are important areas of machine learning that notably deviate from the standard approach. Unfortunately, the focus is still mostly on the latter and that is also the usual point of comparison with the neurosciences. Hence the paper focuses on the standard practice of training from random initialisation.
>
> The mention of shortcut learning is very relevant to what I discuss in the paper. Thanks for pointing that out. I have included a new paragraph in the discussion section about the evidence that classification models learnt spurious features not aligned with perceptual features, as reviewed by Geirhos et al. (2020) and others.

---

### Official Review · AnonReviewer2 · 2020-10-28
**Review of "Rethinking supervised learning: insights from biological learning and from calling it by its name"**

**Rating:** 6
**Confidence:** 3

**Review:**

This paper critiques variants of supervised learning with different names (i.e., semi- and self-supervised learning) and advocates for a different paradigm of learning from data to overcome the limitations of supervised learning. The authors make a case for why semi-supervised learning, self-supervised learning, etc. are indeed forms of supervised learning and not substitutes for supervised learning. As the authors put it, the research community needs to look for something more than different flavors of supervised signals in order to get around the limits of supervised learning. To this end, the authors highlight how supervised learning differs from the type of learning that humans do -- namely via a continuous and coherent stream of inputs instead of an arbitrary sequence of object images. This part can be interpreted as being particularly exciting as it suggests researchers will have to focus on how supervised learning is performed altogether in order to bridge the gap.


Pros:
- The authors recap 'progress' in supervised learning over the past 5-10 years.
- The authors clearly list and describe several discrepancies between current supervised learning and human-like supervised learning that can be useful to the research community at large.

Cons:
- The authors use overstatements (e.g., "... supervised learning went from glory to shame") to describe the general sentiment in the research community, and I don't think it's as severe as they describe.


I suggest the authors re-think their overstatements and rephrase accordingly to better reflect the state of the research community. Otherwise, the authors are pretty clear in their arguments. I haven't read any other critiques of supervised learning or its recent variants, hence this paper provides a novel perspective on what the research community should collectively be doing to bridge supervised learning and human learning. This work has the potential to be quite significant in highlighting what the future of supervised learning should be.

---

> ### Public Comment · ~Alex_Hernandez-Garcia1 · 2020-12-03
> **Brief responses to reviewer's comments**
>
> Thank you for your review. I am glad you had a positive impression of the paper.
>
> Based on your suggestion to rethink the wording of some statements, I have reviewed the manuscript and rewritten some sentences to more accurately express my concerns without overstating them.

---

### Public Comment · ~Alex_Hernandez-Garcia1 · 2020-12-03
**Updated manuscript**

Based on the feedback by the reviewers I have updated the manuscript with some minor changes, hopefully improving the clarity and rigour. I am thankful for the valuable comments of the reviewers.

---

### Decision · Program_Chairs · 2020-11-02

Accept (Poster)